# Mobile Anatomical Total Ankle Arthroplasty—Improvement of Talus Recentralization

**DOI:** 10.3390/jcm10030554

**Published:** 2021-02-02

**Authors:** Faisal Alsayel, Mustafa Alttahir, Massimiliano Mosca, Alexej Barg, Mario Herrera-Pérez, Victor Valderrabano

**Affiliations:** 1Swiss Ortho Center, Schmerzklinik Basel, Swiss Medical Network, Hirschgaesslein 15, 4010 Basel, Switzerland; alsayel002@gmail.com (F.A.); info@drmustafaalttahir.com.au (M.A.); 2King Fahad Specialist Hospital, Dammam, P.O. Box 15215, Dammam 31444, Saudi Arabia; 3Macquarie Limb Reconstruction Unit, Macquarie University Hospital, Sydney, NSW 2109, Australia; 4Department of Orthopaedic and Traumatology, Istituto Ortopedico Rizzoli, Via Dei Colli, 10-40136 Bologna, Italy; massimiliano.mosca@ior.it; 5Department of Orthopaedic, University Medical Center Hamburg-Eppendorf, Martinistraße 52, 20251 Hamburg, Germany; al.barg@uke.de; 6Head Foot and Ankle Unit, Orthopaedic Department, University Hospital of Canary Islands, La Laguna, Carretera Ofra S/N, 38320 San Cristóbal de La Laguna, Santa Cruz de Tenerife, Tenerife, Spain; herrera42@gmail.com

**Keywords:** ankle, ankle osteoarthritis, total ankle arthroplasty, total ankle replacement, vantage

## Abstract

Introduction: Total ankle arthroplasty (TAA) is becoming a more frequent treatment option for end-stage ankle osteoarthritis (OA) as outcomes measures are improving. However, there is concern that malalignment of TAA can result in premature failure of the implant. One of the malalignment issues is the talar sagittal malposition. However, a consensus on the significance of the sagittal translation of the talus in TAA is yet to be established. The aim of this study was, therefore, to clarify whether talus OA subluxation is normalized after the implantation of a mobile TAA. Methods: Forty-nine consecutive patients with symptomatic end-stage ankle OA underwent 50 cementless three-component mobile-bearing VANTAGE TAA with 21 right ankles (42%) and 29 left ankles (58%). Clinical and radiographic outcomes were assessed: Clinical variables: American Orthopedic Foot and Ankle Society (AOFAS) ankle-hindfoot score (0–100), visual analogue scale (VAS, 0–10), and ankle range of motion (ROM). Radiological variables: medial distal tibial articular angle (mDTAA), anterior distal tibial articular angle (aDTAA) and lateral talar station (LTS). Results: The clinical results showed the mean improvement in AOFAS hindfoot score from 42.12 ± SE 2.42 (Range: 9–72) preoperatively, to 96.02 ± SE 0.82 (Range: 78–100) at a mean follow-up of 12 months, with a highly statistically significant difference (*p* < 0.00001). Pain score (VAS) was 6.70 ± SE 0.28 (Range 0–10) preoperatively, and 0.26 ± SE 0.12 (Range: 0–3) at 12-month follow-up, with a highly statistically significant difference (*p* < 0.00001). ROM measurements preoperatively showed a mean of 22.55° ± SE 1.51° (Range: 0–50°), which showed a statistically significant improvement (*p* < 0.0001) to 45.43° ± SE 1.56° (Range: 25–60°) 12 months postoperatively. The radiological analyses revealed the following results: On the coronal view, the mDTAA preoperatively was 88.61 ± SE 0.70 (Range: 78.15–101.10), which improved to 89.46 ± SE 0.40 (Range: 81.95–95.80) at 12 months (not statistically significant—*p* = 0.94). On the sagittal view, the preoperative values of the aDTAA showed 82.66 ± SE 0.84 (Range: 70.35–107.47), which improved to 88.98 ± SE 0.47 (Range: 82.83–96.32) at 12 months postoperatively, with a highly statistically significant difference between preoperative and 12-months values (*p* < 0.00001). The mean LTS values for all patients were 3.95 mm ± SE 0.78 (Range: −11.52 to 13.89) preoperatively and 1.14 mm ± SE 0.63 (Range: −10.76 to 11.75) at 12 months, with a statistically significant difference between preoperative and 12-month follow-up (*p* = 0.01). The review of the radiological TAA osteointegration at 12 months showed no cases of loosening of the implanted TAAs. Two cases (4%) showed a radiolucency and one case (2%) a cyst on the tibial component; no cases had a change on the talar component. No TAA complication/revision surgeries were documented. Conclusion: In the present study, the lateral talar station of anteriorly subluxated ankles showed a significant improvement, i.e., physiological centralization of the talus, in the postoperative period when a mobile-bearing TAA was performed. The anterior/posterior congruency between the talar component and the mobile polyethylene insert of the mobile-bearing VANTAGE TAA allows the sagittal translation of the talus relative to the flat tibial component, reducing the prosthesis strain and failure.

## 1. Introduction

Total ankle arthroplasty (TAA) is becoming more frequent as a treatment option for end-stage ankle osteoarthritis (OA) as outcome measures are improving, particularly when compared to ankle arthrodesis [1,2,3,4]. However, there are still concerns regarding complications of TAA, with studies reporting significant reoperation rates and increasing revision rates [5,6,7]. Specifically, there is concern that malalignment of TAA can result in premature failure of the implant through prosthesis loosening, polyethylene liner wear/fracture/luxation, and prosthetic subsidence [8].

Subluxation of the talus in the setting of end-stage ankle OA is common, either in an anterior or posterior direction [9,10]. In the setting of a normal ankle, the healthy bony anatomy as well as the soft-tissue structures, including joint capsule, ligaments and muscle–tendon units that cross the joint, keep the talus in a physiologically normal position, as well as guide and restrain movement between the talus and the mortise through the entire ankle range of motion [11]. Deviations of talar sagittal position from normal have been attributed to abnormalities of bony shape and alignment as well as soft-tissue constraints, ultimately causing abnormal joint contact pressures and, therefore, ankle OA progression [12]. It is well established that anterior subluxation is the most common deformity in the osteoarthritic ankle, likely as a consequence of an elongated or ruptured anterior talofibular ligament (ATFL) or medial deltoid ligament resulting in chronic ankle instability (CAI) [10,12]. Furthermore, osteoarthritic ankles were associated with weakness in ankle dorsiflexion and plantarflexion, as well as muscle atrophy and significantly reduced electromyographic intensity for the tibialis anterior, gastrocnemius, and peroneal longus muscles [13]. However, knowledge on the significance of the sagittal position and translation of the talus in the setting of a TAA is poor and yet to be established.

Numerous techniques have been described in the literature to measure talar sagittal position, with the methods aimed at quantifying the anteroposterior tibial-talar orientation without using ankle joint line radiographic landmarks, which also allows their use in the setting of TAA. The tibial-axis-to-talus ratio [14], the posterior-tibial-line-to-talus ratio [14], the tibial-axis-to-lateral-process distance [14], the contact-point ratio [15], the talar-centre-of-rotation-to-tibial-axis distance [12], and the lateral talar station (LTS) [16] were evaluated for inclusion as the radiographic analysis technique. The defined parameters of malalignment and subluxation in the setting of end-stage ankle OA cannot necessarily be directly extrapolated to the setting of TAA. Presently, there are no defined criteria for outliers of normal sagittal talar position in TAA.

The aim of this study was to clarify whether anterior subluxation in the setting of end-stage ankle OA is indeed a concerning feature that requires special attention in performing TAA, and whether talar subluxation is normalize in postoperative radiographs after implantation of a mobile three-component TAA.

## 2. Methods

### 2.1. Study Design

A prospectively collected database of patients undergoing a three-component mobile TAA at the senior author’s institution was used to identify a consecutive series of patients. This study was approved by the local ethical institutional review Board (IRB) and informed consent was obtained from all patients.

Inclusion criteria were the following:
Consecutive endstage painful ankle OA patients treated with cementless three-component mobile-bearing VANTAGE Total Ankle Arthroplasty (TAA; Exactech, Gainesville, FL, USA; The 5th generation total ankle arthroplasty system [17]) (Figure 1).Minimum of 12 months follow-up with clinical and radiographic evaluation.

Due to the prospectively collected database, no patients were excluded. The following variables were registered:

Clinical variables: American Orthopedic Foot and Ankle Society (AOFAS) ankle-hindfoot scale (0–100), visual analogue scale (VAS, 0–10), and ankle range of motion (ROM).

Radiological variables: medial distal tibial articular angle (mDTAA), anterior distal tibial articular angle (aDTAA) and lateral talar station (LTS).

### 2.2. Patients

Fifty consecutive TAAs with 21 right ankles (42%) and 29 left ankles (58%) were performed by a single surgeon in 49 patients with symptomatic end-stage ankle OA: 28 (57%) females, 21 (43%) males, with a mean age of 60.1 years (range, 23–81 years). The diagnoses included post-traumatic ankle OA in 38 cases (76%), primary ankle OA in 7 cases (14%), and secondary ankle OA in 5 cases (10%) (Table 1).

### 2.3. Surgical Technique and Post-Operative Protocol

A standard anterior approach to the ankle using the interval between tibialis anterior and extensor hallucis longus tendon was utilised for the implantation of the VANTAGE TAA. Standardized surgical technique was used according to Valderrabano et al. and the manufacturers’ instructions [17]. The tibial and talar cuts and prosthesis preparation and placement was guided by image intensifier throughout the procedure, with strict attention to distal tibial slope, talar sagittal positioning, and avoiding overstuffing of the joint. The tibial resection was based on the physiological tibial alignment. The talar preparation (cuts and pegs) was then performed by anatomical placement at the centre of the talus bone, with support of the fluoroscan. The tibia was finalized by preparing the tibial cage and three tibial antirotation pegs holes for the tibial component. Trial implantation indicates the right position, size of the components, stability and range of motion. The primary three components mobile VANTAGE TAA were implanted. Necessary additional procedures were performed in 36 patients (70%) during the same ankle replacement procedure based on preoperative and intraoperative evaluation. Routine postoperative leg elevation and physiotherapy, focusing on lymphatic drainage, Achilles stretching and ROM exercises, was instituted. Immediate mobilization with a walker and crutches. Immediate full weight bearing was only allowed if there were no additional bony procedures performed, such as fusions or osteotomies. Alternatively, the patients were advised to limit weightbearing to 15 kg for 6 weeks if a concomitant procedure required restriction of weightbearing.

### 2.4. Clinical Evaluation

All patients were evaluated preoperatively and at their postoperative 12-month follow-up clinically using the American Orthopedic Foot and Ankle Society (AOFAS) ankle-hindfoot score (0–100 points), pain visual analogue scale (VAS, 0–10 points), and ankle range of motion (ROM) as sum of ankle dorsiflexion and plantarflexion. Data regarding perioperative complications and additional procedures were also obtained.

### 2.5. Radiographic Evaluation

Routinely, standardised weightbearing radiographic views (ankle ap, foot lateral, foot ap, Saltzman Hindfoot view) were taken preoperatively, immediately 6 weeks postoperatively, at 3 months, 6 months, 1 year, and then annually thereafter. For the purpose of this study, we analysed the preoperative, 6-month and 12-month postoperative radiographs to judge the change of TAA centre of rotation over time. For the radiological analysis and measurements, the INFINITT PACS (INFINITT Healthcare Co. Ltd., Seoul, Korea) software was used. Plain radiographs were evaluated by two independent observers who were not involved in the surgical and clinical treatment of the patients, to minimise potential bias and increase inter-observer reliability.

The medial distal tibial articular angle (mDTAA, degrees) [18] on the ankle OA as well as on the TAA as medial tibial component coronal angle were used to identify coronal alignment, measured on the anteroposterior radiograph as the medial angle subtended by the anatomic axis of the tibia and the tibial plafond line or inferior border of the tibial component, with positive values equal to valgus result and negative values equal to varus result (Figure 2A,B). The sagittal alignment assessed was by the anterior distal tibial articular angle (aDTAA, degrees) [19] on the ankle OA and on the TAA as anterior tibial baseplate angle [14,20] on the lateral radiograph, measured as the anterior angle formed from the anatomic axis of the tibia and the line connecting the distal points on the anterior and posterior tibial articular surface or inferior border of the tibial component (Figure 2C,D).

Among the methods listed above, we utilised the LTS (Lateral Talar Station, mm) technique described in the literature as the most reproducible and least sensitive to sagittal ankle position, with defined normal ranges, to quantify sagittal talar position. The LTS method as described by Veljkovic et al. [16]—a modification of the previously described TibCOR measurement by Magerkurth et al. [12]—measures the distance between the tibial long axis, defined as a line joining the centre of two circles in the tibia (10 cm and 5 cm from tibial plafond), and a perpendicular line extending to the centre of rotation of the talus (as an average of the two condylar centres of rotation), reported in millimetres. Values were defined as high if the centre of the circle was anterior to the tibial axis (>3.1496 mm) and low if lying posterior to the tibial axis (<−0.8076 mm) [16] (Figure 2E,F). In order to get more insight into the LTS measurement, the following subgroups analyses were performed: LTS (overall total values), LTS-A: preoperatively anteriorly subluxated ankle OA cases, LTS-P: preoperatively posteriorly subluxated ankle OA cases, LTS-N: preoperatively non-subluxated ankle OA cases.

Radiolucency was assessed and defined as a gap less than 2 mm wide between the implant and the bone, not represented on the initial 6 weeks postoperative radiographs [21]. Osteolysis was defined as a demarcated non-linear lytic lesion of a width of 2 mm or more [21]. The radiological zones of the VANTAGE Total Ankle System were the zones dividing the area around the tibial component into four vertical segments on the AP and lateral views [22]. The talar component was split into three separate segments. Possible heterotopic ankle ossification was graded based on the classification of heterotopic ossification of the ankle following TAA reported by Lee et al. [23].

### 2.6. Statistical Analysis

Descriptive statistics were employed to assess the means, standard deviations, standard error and confidence intervals for each measure. Repeated-measures analysis of variance (ANOVA) was used to determine significant differences among preoperative, 6-month and 12-month postoperative follow-up time points for each continuous dependent variable. The variables of interest were clinical measures (AOFAS, VAS, and ROM) and radiographic component measures including mDTAA, aDTAA, and LTS. Subgroup analyses were performed on preoperative anteriorly translated, neutral and posteriorly translated talar groups. In addition, the number of patients with complications was analysed as defined by Veljkovic et al. [6]. Linear regression analyses were utilised to determine any relationship between variables. The statistical significance level was set at *p* < 0.05.

## 3. Results

A review of the clinical results showed the mean improvement in AOFAS anklehindfoot score(points) from 42.12 ± SE 2.42 (range: 9–72) preoperatively, to 96.02 ± SE 0.82 (range: 78–100) at a mean follow-up of 12 months, with a highly statistically significant difference (*p* < 0.00001). Pain score VAS (points) was 6.70 ± SE 0.28 (range 0–10) preoperatively, and 0.26 ± SE 0.12 (range: 0–3) at 12-month follow-up, with a highly statistically significant difference (*p* < 0.00001). Ankle ROM measurements (degrees; sum dorsiflexion and plantarflexion) pre-operatively showed a mean of 22.55° ± SE 1.51° (range: 0–50°), which showed a statistically significant improvement (*p* < 0.0001) to 45.43° ± SE 1.56° (range: 25–60°) at 12 months postoperatively (Table 2).

The radiological analyses revealed the following results: On the coronal view, the medial distal tibial articular angle (mDTAA, degrees) preoperatively was 88.61 ± SE 0.70 (range: 78.15–101.10), which improved to 89.46 ± SE 0.40 (range: 81.95–95.80) at 12 months (not statistically significant—*p* = 0.94). On the sagittal view, the preoperative values of the anterior distal tibial articular angle (aDTAA, degrees) showed 82.66 ± SE 0.84 (range: 70.35–107.47), which improved to 88.98 ± SE 0.47 (range: 82.83–96.32) 12 months postoperatively, with a highly statistically significant difference between preoperative and 12-month values (*p* < 0.00001).

The mean LTS (Lateral Talar Station, mm) values for all patients were 3.95 mm ± SE 0.78 (range: −11.52 to 13.89) preoperatively and 1.14 mm ± SE 0.63 (range: −10.76 to 11.75) at 12 months, with a statistically significant difference between preoperative and 12-month follow-up (*p* = 0.01). The mean LTS for 27 TAA with anterior subluxation (LTS-A) preoperatively was 7.99 mm ± SE 0.54 (range: 3.30–13.89) and 1.59 mm ± SE 0.71 (range: −5.72 to 11.75) at 12 months postoperatively, with a highly statistically significant difference between preoperative and 12-month values (*p* < 0.00001). The mean LTS for 16 TAA with no subluxation (LTS-N) preoperatively was 1.22 ± SD 1.02 (range: −0.63 to 2.70) and 2.01 ± SD 4.82 (range: −10.67 to 10.48) at 12 months postoperatively, with no statistically significant difference between preoperative and 12-month values (*p* = 0.53). The mean LTS for 7 TAA with posterior subluxation (LTS-P) preoperatively was −5.44 ± SD 3.63 (range: −11.52 to −2.17) and −2.55 ± SD 5.13 (range: −10.76 to 3.20) at 12 months postoperatively with, no statistically significant difference between preoperative and 12-month values (*p* = 0.25) (Table 3) (Figure 3A,B).

The review of the radiological TAA osteointegration at 12 months showed no cases of loosening of the implanted VANTAGE TAAs. Two cases (4%) showed 2 mm of tibial radiolucency (1 case (2%) posterior/lateral with a medial malleolar stress fracture requiring subsequent medial malleolus ORIF without TAA revision surgery, and 1 case (2%) anterior/lateral). Both tibial radiolucency cases showed following radiological measurements: Case 1: mDTAA (83.45), aDTAA (85.96), and LTS (−2.14), Case 2: mDTAA (88.53), aDTAA (90.02), and LTS (0.22). One case (2%) showed an anterior tibial cyst with no TAA loosening at 12 months. This tibial component cyst case had following radiological values: mDTAA (88.14), aDTAA (91.09), and LTS (−2.08). Regarding the talar TAA component, no cases of talar loosening, cyst formation or osteolysis were documented at 12-month follow-up. The radiological assessment of periarticular ossification revealed seven cases in total (8%), all of whom were asymptomatic: six cases (12%) of grade 1, one case (2%) of grade 2, and no cases of grade 3 or 4 changes as per the classification devised by Lee et al. [23].

In the linear regression analysis for correlation between clinical and radiological scores, there were no significant predictors of AOFAS score or ROM scores compared to postoperative radiological measures, with low coefficients of determination (R2) throughout: AOFAS vs. aDTA: Multiple R = 0.46; R2 = 0.21; *p* < 0.000001; AOFAS vs. LTS: Multiple R = 0.29; R2 = 0.085; *p* = 0.004; ROM vs. LTS: Multiple R = 0.27; R2 = 0.077; *p* = 0.006; AOFAS vs. mDTA: Multiple R = 0.16; R2 = 0.026; *p* = 0.119.

No TAA complication/revision surgeries were documented in all the implanted TAAs.

## 4. Discussion

The most important finding of this study demonstrated that a mobile bearing TAA restores an anteriorly subluxated talus to an anatomical position with a highly statistically significant difference (*p* < 0.00001). The majority of patients in this study had an anteriorly translated talus (LTS-A: 27 cases of 50 cases; 54%). In addition, the improvement in clinical outcome scores was also highly statistically significant, demonstrating that the VANTAGE TAA provided significantly improved clinical results across outcomes scores, as AOFAS ankle-hindfoot scores, pain scores, and ankle ROM scores.

Proper alignment is important for TAA longevity [24,25]. The purpose of the current study was to assess the repositioning of an anteriorly subluxated talus into the mortise after a mobile-bearing TAA, and the effect on clinical and radiologic outcomes of mobile bearing TAA. The normalisation of lateral talar position/station should theoretically decrease wear and increase longevity of the prosthesis by minimizing shear forces at the bone-prosthesis interface [26,27,28].

There is some concern that anterior translation of the talus in the setting of TAA can result in edge loading of the polyethylene liner, and lead to early wear and premature failure of the TAA [8]. Younger et al. [29] highlighted it as a pathological phenomenon, and set out to describe the numerous causes for anterior translation as issues that need addressing, particularly increasing the reoperation and failure rate of TAA. The underlying causes of anterior translation are outlined as numerous, including (1) unaddressed tightness of the posterior capsule, osteophytes and other structures causing a posterior contracture, (2) overstuffing of the TAA with polyethylene, (3) increased anterior slope of the tibial component, (4) anteriorisation of the talar component on the talus in the index procedure, or (5) migration of any of the components of a TAA. The present study reveals the scientific evidence that the VANTAGE mobile three component TAA is able to compensate pathological sagittal mal position of the talus, avoiding this way a prosthetic failure.

However, Lee et al. [24] in a comparison of 120 ankles undergoing a cementless three-component TAA, with preoperative radiographically measured anteriorly translated talus in 50 ankles versus non-translated talus in 54 ankles, demonstrated no significant difference in clinical outcomes or range of motion measurements between the two groups, with 92% of the anteriorly translated ankles showing relocation at 6 months radiographically, progressing to 96% relocation at 12 months. There was no evidence of increased complication rate between the two groups at a mean follow-up of 42.8 months.

In the present setting of the mobile-bearing VANTAGE TAA, the talar centre of rotation changed from its initial position during the direct postoperative period as the surrounding soft tissues adjusted to the new alignment and motion resulting from the TAA. This could be explained by improved muscle function (torque and electromyographic intensity) after TAA, as described by Valderrabano et al. [13] in their prospective study of muscle rehabilitation in TAA for unilateral severe ankle OA. At 12 months postoperatively, the dorsiflexion and plantarflexion torque, as well as the mean EMG intensity, improved and were close to the level of the contralateral healthy leg.

Barg et al. [30], in their retrospective cohort study of 317 patients who underwent a Hintegra mobile-bearing TAA (Newdeal, Lyon, France/Integra, Plainsboro, New Jersey), reported statistically significant improvements in postoperative pain, AOFAS hindfoot scores, and ankle ROM in the patients with Antero-Posterior Offset Ratio of 0 (127 ankles; 34.5%), when compared with either the positive- or negative-offset groups. The results strongly support the assumption that proper positioning of the talar component in the sagittal plane results in better postoperative pain relief and functional outcome.

Valderrabano et al. [25] reported, in their cohort of 68 TAA whom underwent a S.T.A.R. prosthesis (LINK, S.T.A.R. Scandinavian Total Ankle Replacement, Waldemar Link, Hamburg, Germany now by Stryker, Kalamazoo, MI, USA) with 3.7 years follow-up, the prosthetic talar centre-of-rotation converged toward the longitudinal tibial axis, on average 0.7 mm (*p* > 0.05). The distance between the prosthetic talar centre-of-rotation and the tibial longitudinal axis was measured, with an overall mean distance of 1.6 mm anterior to the tibial longitudinal axis on the lateral neutral weight bearing radiographs at last follow-up, compared to 2.3 mm in direct postoperative radiographs.

Usuelli et al. [26], in their case series of 66 consecutive patients who underwent TAA with the mobile-bearing Hintegra prosthesis, demonstrated a significant increase in the tibio-talar ratio from 2 to 6 months postoperatively (34.6–37.2%). In their cohort, significant movement of the talus occurs within the first 6 months postoperatively. This may be explained by the rebalancing of muscle and ligament forces after surgery.

Additionally, Usuelli et al. [27] found a significant difference in the antero-posterior translation of the talus component over time between unconstrained, mobile-bearing TAA (Hintegra) and a semi-constrained, fixed-bearing TAA (Zimmer TM). In their cohort, they compared 71 consecutive ankles that received a mobile-bearing TAA and 24 ankles that received the fixed-bearing TAA. At 6 months postoperatively, there was a significant increase in the tibio-talar ratio from 2 to 6 months in the mobile-bearing TAA (*p* < 0.001), representing a normalisation of sagittal talar position. The changes in the tibio-talar ratio were not significant in the fixed-bearing TAA group. These results demonstrate that the mobile-bearing prosthesis design most replicated the normal ankle biomechanics in restoring sagittal talar position.

Wood et al. [31] evaluated the effect of tibial inclination on the anteroposterior position of the talus in the mortise in his cohort of 200 TAA with the Scandinavian Total Ankle Replacement (STAR). In cases when the anterior distal tibial angle (β-angle) was <83° (anteriorly inclined), there was anterior talar translation with statistical significance. The talus translated posteriorly when the β-angle was 83° to 90° (normal inclination) or >90° (posterior inclination), although not statistically significant.

To achieve neutral coronal alignment of a TAA, it is important to implant the TAA correctly (physiological angles) and to address possible hindfoot mal deformities through additional procedures. Zafar et al. [32] showed in their cohort of 322 Hintegra TAA with 12 years follow up, a reduction in the risk of revision for each increased degree of the postoperatively mDTA, suggesting that valgus alignment of TAA is safer than varus alignment.

In our series, serial radiographic follow-up did not show increased osteolysis or loosening, even though the tibial component showed focal radiolucency in two cases and focal cystic osteolysis in one case, all three cases without TAA loosening. Such tibial changes are considered as benign findings because of tibial stress shielding [11]. The talar component showed no osteolysis or loosening in our cohort.

The strength of the study is that functional outcomes, including AOFAS ankle-hindfoot scores, VAS for pain, and ankle ROM measurements, were evaluated for all patients, and follow-up radiographic data was available for all patients.

There were limitations to the present cohort study. First, it was a single institutional database and all the cases performed by one highly experienced surgeon. The postoperative follow-up was short in duration. However, as in other studies, the results demonstrated the greatest improvement of the talar position occurred in the first 6 to 12 months postoperatively [16,25,26,27]. Whether that phenomenon results in improved survivorship of the TAA will require longer-term follow-up and outcome reviews. It is the aim of the authors to continue for longer-term follow-ups.

Second, the study is a review of a mobile-bearing TAA with no fixed-bearing TAA comparison. Further studies with larger cohorts of participants, comparing both mobile- and fixed-bearing groups, and longer follow-up are needed to confirm these short-term clinical and radiological data. Additionally, the sample size in the neutrally and posteriorly translated groups were relatively small, owing to the lower incidence of both talar stations in the setting of ankle OA.

## 5. Conclusions

In the present study, the radiological lateral talar station (LTS) of anteriorly subluxated ankles showed a significant improvement, i.e., physiological centralization of the talus, in the postoperative period when a mobile-bearing TAA was performed. The unique design of the talar component (which mimics a patient’s native anatomy), as well as anterior/posterior congruency between the talar component and the polyethylene insert of the mobile-bearing VANTAGE TAA allows the sagittal translation of the talus relative to the flat tibial component, resulting in the recentralization of the talus as the surrounding soft tissues adjust to the new alignment and motion over time.

However, careful attention should be paid to the distal tibial and talar sagittal component positioning, and avoiding overstuffing of the joint. The early clinical and radiographic results of the VANTAGE TAA are of good quality. Longer-term results for this mobile-bearing TAA are under investigation.

## Figures and Tables

**Figure 1 jcm-10-00554-f001:**
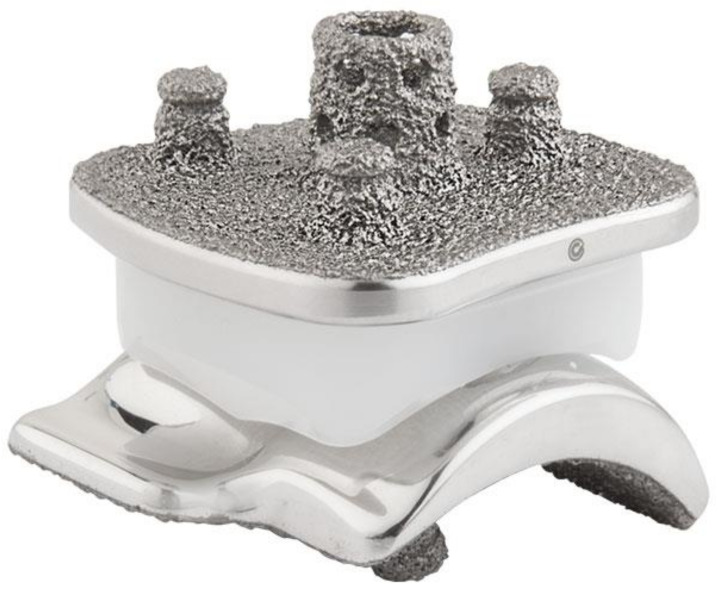
VANTAGE Mobile Bearing Total Ankle Arthroplasty.

**Figure 2 jcm-10-00554-f002:**
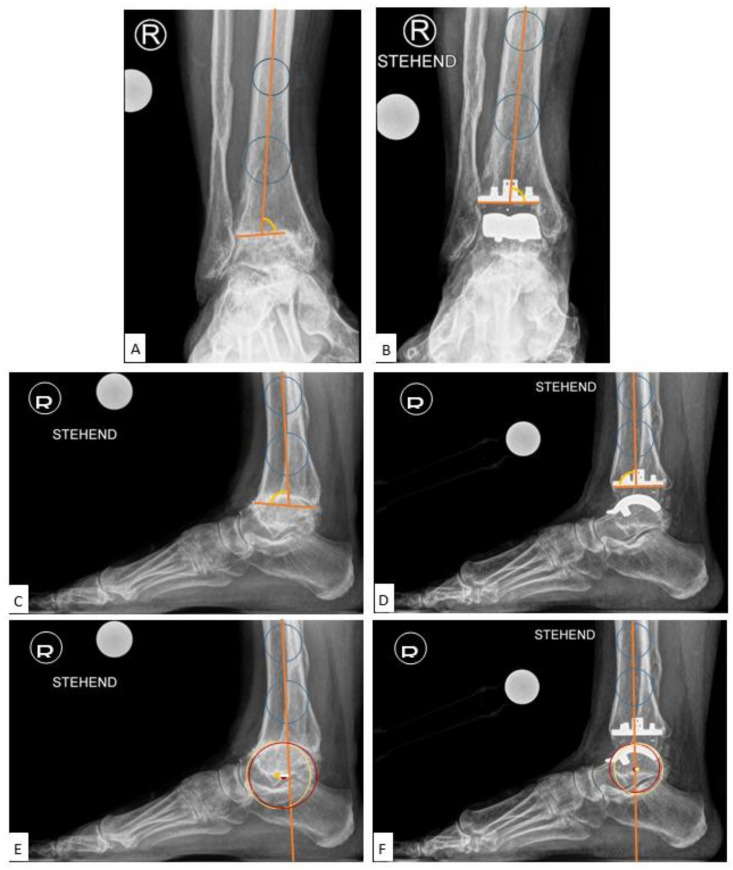
Preoperative and Postoperative Radiological Measurements. (**A**): Medial distal tibial articular angle (mDTAA) on the ankle osteoarthritis (OA), (**B**): Medial distal tibial articular angle (mDTAA) on the total ankle arthroplasty (TAA) as medial tibial component coronal angle, (**C**): Anterior distal tibial articular angle (aDTAA) on the ankle OA, (**D**): Anterior distal tibial articular angle (aDTAA) on the TAA as anterior tibial baseplate angle. Lateral talar station (LTS). (**E**): Preoperative on the ankle OA, (**F**): Postoperative on the TAA.

**Figure 3 jcm-10-00554-f003:**
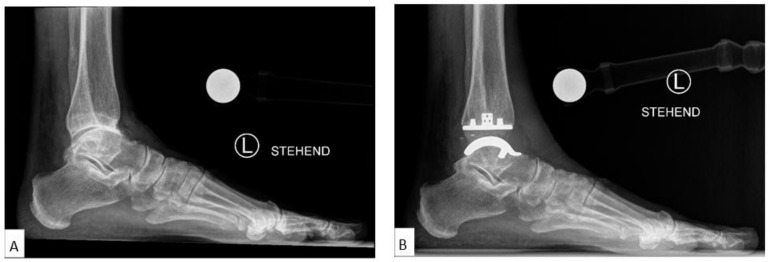
Case with Centralization of Talus after Implantation of VANTAGE Mobile Bearing Total Ankle Arthroplasty. The talar centre of rotation/Lateral Talar Station (LTS) converged toward the longitudinal tibial axis from 10 mm preoperatively (**A**) to 2 mm postoperatively in 12 months follow-up radiograph (**B**).

**Table 1 jcm-10-00554-t001:** Demographics.

	Results
Number: Cases, Patients	50, 49
Patients’ Mean Age: years (range)	60.1 (23–81)
Gender: Female, Male	28 (57%), 21 (43%)
Aetiology: Posttraumatic Ankle OA (*n* (%))	38 (76%)
Primary Ankle OA (*n* (%))	7 (14%)
Secondary Ankle OA (*n* (%))	5 (10%)

OA: Osteoarthritis.

**Table 2 jcm-10-00554-t002:** Clinical Results.

Clinical Variables	Ankle OA Preoperative	TAA 12 Months Postoperative	Difference Significance (*p*-Value)
AOFAS Score	42.12 ± 2.42 (9–72)	96.02 ± 0.82 (78–100)	<0.00001
Pain (VAS)	6.70 ± 0.28 (0–10)	0.26 ± 0.12 (0–3)	<0.00001
ROM	22.55 ± 1.51 (0–50)	45.43 ± 1.56 (25–60)	<0.00001

All results expressed in Mean ± Standard Error (Range: Minimum–Maximum). OA: Osteoarthritis. TAA: Total Ankle Arthroplasty. AOFAS: American Orthopaedic Foot and Ankle Society Ankle-Hindfoot Score. VAS: Visual Analogue Scale 0–10. ROM: Range of Motion, of sum of Ankle Dorsiflexion and Plantarflexion.

**Table 3 jcm-10-00554-t003:** Radiological Measurements.

Variables	Ankle OA Preoperative	TAA 6-Months Postoperative	Difference Significance (*p*-Value) ^1^	TAA 12-Months Postoperative	Difference Significance (*p*-Value) ^2^
mDTAA	88.61 ± 0.70 (78.15–101.10)	89.42 ± 2.63 (82.64–94.82)	0.30	89.46 ± 0.40 (81.95–95.80)	0.94
aDTAA	82.66 ± 0.84 (70.35–107.47)	89.07 ± 3.14 (83.19–96.44)	<0.00001	88.98 ± 0.47 (82.83–96.32)	<0.00001
LTS	3.95 ± 0.78 (−11.52–13.89)	1.37 ± 0.63 (−10.18–12.26)	0.01	1.14 ± 0.63 (−10.76–11.75)	0.01
LTS-A *	7.99 ± 0.54 (3.30–13.89)	1.97 ± 0.75 (−5.16–12.26)	<0.00001	1.59 ± 0.71 (−5.72–11.75)	<0.00001
LTS-P ^†^	−5.44 ± 1.37 (−11.52–−2.17)	−2.75 ± 1.76 (−10.18–3.32)	0.25	−2.55 ± 1.94 (−10.76–3.20)	0.25
LTS-N ^‡^	1.22 ± 0.25 (−0.63–2.70)	2.21 ± 1.14 (−9.19–10.48)	0.40	2.01 ± 1.20 (−10.67–10.48)	0.53

All results expressed in Mean ± Standard Error (range, Minimum–Maximum). OA: Osteoarthritis. TAA: Total Ankle Arthroplasty. ^1^: Significance of change from preoperative measurements to 6-months measurements expressed as *p*-value; Significance set at <0.05. ^2^: Significance of change from preoperative measurements to 12-months measurements expressed as *p*-value; Significance set at <0.05. mDTAA: Medial distal tibial articular angle. aDTAA: Anterior distal tibial articular angle. LTS: Lateral Talar Station, overall values: *n* = 50 cases. * Pre-operatively Anteriorly subluxed Lateral Talar Station subgroup (> 3.1496 mm): *n* = 27 cases. ^†^ Pre-operatively Posteriorly subluxed Lateral Talar Station subgroup (<–0.8076 mm): *n* = 7. ^‡^ Pre-operatively Non-subluxed Lateral Talar Station subgroup (–0.8076 mm to 3.1496 mm): *n* = 16.

## Data Availability

The data presented in this study are available on request from the corresponding author. The data are not publicly available for ethical and privacy reasons.

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
