# Peer review of "Mobile Anatomical Total Ankle Arthroplasty—Improvement of Talus Recentralization"

_jcm, 2021, doi:10.3390/jcm10030554_

Round 1
Reviewer 1 Report
Dear authors,
thank you for submitting your work to Journal of Clinical Medicine. My reviewer comments are as follows:
Overall, this study is well written and has value in the number of implants done, although performed by the design surgeon.
Abstract, Introduction, Methods: the writing is clear and the Method section is appropriate for the aim of the study. Please provide details about the surgical procedures that were performed concomitant with TAA.
Please provide the reference for the definition of radiolucency and osteolysis. How did you assess, ankle ROM? Please clarify.
In table 1, I recognized a young adult included in the study group. 23 years for a TAA is VERY young. Is that an outlier or do you routinely do TAA in a young age group, that number makes me want to see a more detailed breakdown of patient age and etiology of arthritis.
Results: This section contains too much data. The table 3 is complete, therefore in the text provide the pre-op and last follow up data about radiographic measurements.
In this section, I cannot find the results of the linear regression analysis. Please, it’s very important to report the association between clinical outcomes and radiological parameters.
Did you analyze clinical outcomes considering the additional procedures?
Discussion: Please also discuss the differences of reported outcomes by implant inventors, compared with outcomes published by other users and those shown in registry data. Most of the Authors cited in the discussion declare conflict of interest with some TAA system.
The last Author is the designer of the implant, please provide a statement about the potential bias associated with this condition.
Conclusions: These conclusions need to be softened, modified in order to reflect only the study findings.
Author Response
Thanks for your valuable comment and feed back
we addressed the comments that you mention which improve our paper
the answer was attached in pdf file

Reviewer 2 Report
Very interesting article.
I have some minor comments:
Line 116 - no need for decimals, just use 60 years.
In Methods . since the Vantage system probably is not well known by the Foot and Ankle Community it would be appropriate to just add e sentence refering to reference 19 regarding more information about the prosthesis.. By the way reference 19 is now published, it should be 2020!
Line 179 - should be Table 3.
Results - the first 3 paragraphs are very difficult to read, numbers, numbers and numbers…..All this information is clearly stated in table 3. Much easier for a reader to minimize the text and refer to the table.
Diskussion - if you use AOFAS, that nowadays is proposed to be abandoned, it would be proper to discuss the shortcomings of this score. Pinsker-Daniels - Foot Ankle Int 2011;32(9):841 and Macaulay et al. Foot Ankle Spec 2018;11(5):416.
Reference 3 - Should be: …..Carlsson Å. 10-year survival……. the other words in this reference are very odd to use in this context.
There are some inconsistencies in the reference list, some have first and last pages, som just the first page
Author Response
Thanks for your valuable comments and feedback. We addressed the inquiries that raised by the reviewer.
Line 116 - no need for decimals, just use 60 years.
Revised
In Methods . since the Vantage system probably is not well known by the Foot and Ankle Community it would be appropriate to just add e sentence refering to reference 19 regarding more information about the prosthesis.. By the way reference 19 is now published, it should be 2020!
Done and the year of the application corrected
Line 179 - should be Table 3.
Done and corrected
Results - the first 3 paragraphs are very difficult to read, numbers, numbers and numbers…..All this information is clearly stated in table 3. Much easier for a reader to minimize the text and refer to the table.
We can rewrite the result as you request by adding only the preop and last follow up radiological value. However, for higher scientific writing issues we decide to leave the text as it is. This way, the readers can get into the real detail.
Discussion - if you use AOFAS, that nowadays is proposed to be abandoned, it would be proper to discuss the shortcomings of this score. Pinsker-Daniels - Foot Ankle Int 2011;32(9):841 and Macaulay et al. Foot Ankle Spec 2018;11(5):416.
Although in the literature there is a controversy about the AOFAS score, the AOFAS society itself and many recent studies are using it, as it is important to be able to compare with other old and recent studies.
Reference 3 - Should be: …..Carlsson Å. 10-year survival……. the other words in this reference are very odd to use in this context.
Done and corrected
There are some inconsistencies in the reference list, some have first and last pages, som just the first page
Done and corrected